# Characterizing a Year-Round Particulate Matter Concentration and Variation under Different Environmental Controls in a Naturally Ventilated Dairy Barn



Yujian Lu [1,2], Xiao Yang [1,2], Lei E [1,2], Zhiwei Fang [1,2], Yongzhen Li [1,2], Chao Liang [1,2], Zhengxiang Shi [1,2] and Chaoyuan Wang [1,2,*]

1 Department of Agricultural Structure and Bioenvironmental Engineering, College of Water Resources and Civil Engineering, China Agricultural University, Beijing 100083, China; luyujian@cau.edu.cn (Y.L.); yongzhenli@cau.edu.cn (Y.L.)
2 Key Laboratory of Agricultural Engineering in Structure and Environment, Ministry of Agriculture and Rural Affairs, Beijing 100083, China
* Correspondence: gotowchy@cau.edu.cn

**Abstract:** A mixing fan and spraying system is commonly used to control the indoor environment of naturally ventilated dairy barns worldwide. However, its impact on particulate matter (PM) concentration and variation is still unclear due to the lack of year-round field data. To systematically characterize the PM dynamics under different environmental controls (namely, EC1: No Fans and No Spraying; EC2: Fans; EC3: Fans and Spraying), a year-round continuous monitoring of PM less than 2.5 μm in aerodynamic diameter ($PM_{2.5}$) and total suspended particle (TSP) concentrations, as well as indoor environmental factors, was carried out inside a naturally ventilated dairy barn using an IoT-based sensor monitoring network. Results showed that the hourly mean TSP and $PM_{2.5}$ concentrations were 94.7 μg m$^{-3}$ and 49.8 μg m$^{-3}$, respectively. EC2 had a higher TSP content (116.6 μg m$^{-3}$) than EC1 (98.0 μg m$^{-3}$) and EC3 (81.9 μg m$^{-3}$). EC1 had the greatest $PM_{2.5}$ concentration (57.1 μg m$^{-3}$), followed by EC2 (48.3 μg m$^{-3}$) and EC3 (44.7 μg m$^{-3}$). EC1 showed clear TSP and $PM_{2.5}$ fluctuations during the daily operations at 07:00 to 08:00 and 18:00 to 19:00, while irregular peaks in EC2 and a relatively steady diurnal variation in EC3 were found. Daily Tsp concentrations in the three ECs did not exceed 300 μg m$^{-3}$. However, 17.8%, 11.5%, and 4.8% of the observed days in EC1, EC2, and EC3 had daily mean $PM_{2.5}$ concentrations above the healthy threshold (75 μg m$^{-3}$), mostly from 07:00 to 08:00 and 22:00–07:00. In conclusion, the mixing fan and spraying system had significant effects on PM concentration and variation, and more protection procedures should be taken for farm workers to prevent long-term health risk exposure, to EC1 in particular.

**Keywords:** environmental control; IoT; mixing fans; particulate matter; spraying system

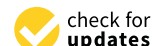



## 1. Introduction

Particulate matter (PM) generation and emission from dairy farming have a potential effect on the health and welfare of the animals [1], farm workers [2] and even neighbors of a dairy farm [3]. The PM with gaseous pollutants, bacteria, and viruses can compromise an animal's and human's respiratory health, causing chronic cough and/or phlegm, chronic bronchitis, allergic reactions, and asthma-like symptoms [1]. Previous studies on dairy operations found that the average PM concentrations at dairy farms were above the postulated threshold for respiratory effects [4,5] and contributed to the burden of respiratory disease among their neighbors [6,7]. A twelve-year study on the respiratory status of dairy farmers suggested that respiratory function impairment correlated with cumulated exposure to organic dust [8]. With the expansion of the scale and number of dairy facilities,

health and environmental issues initiated by PM of dairy farming are attracting more public concerns.

Limited studies have been conducted regarding PM generation and emissions in dairy farming. Most of them documented $PM_{2.5}$ and total suspended particulate (TSP) concentrations and emission levels during the measurements over the past decades [9,10]. Based on a three-month measurement of large dairy facilities with naturally ventilated systems, Zhao et al. found that the average TSP concentrations ranged from 0.9 to 1.5 mg m$^{-3}$ [11]. Others focused on distributions of indoor PM concentration and their influencers, such as temperature (T), relative humidity (RH), carbon dioxide ($CO_2$), and ammonia ($NH_3$) concentrations [12,13]. Joo et al. investigated the relationship between PM concentrations and cow activity [14]. However, the impact of environmental controls (ECs) on PM characteristics inside naturally ventilated dairy barns has been scarcely reported, which is critical to environmental regulations and health assessment. ECs were found to significantly impact PM characteristics for pig and poultry farming with mechanically ventilated barns [15,16].

Fans and spraying cooling systems are typically provided in dairy buildings and are currently becoming a standardized EC method worldwide [17]. By creating an appropriate interior microenvironment, fans and spraying cooling systems have been shown to be effective in improving dairy cows' comfort and milk production [18,19]. Their operation also affects PM concentration and variation by increasing indoor air velocity and RH. Spraying systems in livestock were found to reduce air temperature by 3.0–10.5 °C and increase RH by 13.7–20.0% [20]. Previous studies have shown that flow speed and humidity affect indoor PM deposition and cause structural changes in PM [21,22]. For example, spraying systems are used for dust suppression during the mining process [23]. Furthermore, in practice, fans and spraying systems are commonly controlled at three different modes, namely EC1 (No Fans and No Spraying), EC2 (Fans), and EC3 (Fans and Spraying) in cold, mild, and hot seasons, respectively. However, the effect of different ECs on PM characteristics in naturally ventilated dairy barns remains unknown due to a lack of sufficient field data, especially for those covering an entire production cycle. Since year-round monitoring with laboratory-based equipment was costly, time-consuming, and labor-intensive, the majority of PM measurements inside the dairy building were conducted for only a couple of days in one or several seasons [9–13].

An intermittent sampling strategy of PM concentrations might not be able to fully illustrate the ECs impact on PM due to the temporal and spatial variations of PM and complicated influencers. For example, Joo's study demonstrated distinct seasonal and diurnal changes in PM concentration in a naturally ventilated dairy building [14]. Furthermore, Fang's measurement revealed a spatial discrepancy between PM concentrations at the roof and cubicle levels [24]. Thus, in order to acquire accurate PM data under varied ECs, year-round and continuous monitoring at multiple points is assumed to be the most reliable method to study the environment of naturally ventilated dairy barns. Li et al. systematically evaluated the health risk of PM concentration based on continuous monitoring of PM throughout the annual production cycle of laying hen farms and suggested protective measures were required to prevent the respiratory tract from irritation or infection during the working period [25]. In recent times, as the detecting sensors advance towards miniaturization with great cost drop, they provide a practical and economical method to build an online monitoring network using Internet of Things (IoT) technology to carry out a year-round assessment of the animal environments [26,27], such as odor and hazardous gas monitoring inside a swiftlet farming [28] and PM distribution regularity measurement in an aviary house [29].

The objectives of this study were to systematically characterize the year-round $PM_{2.5}$ and TSP concentration and variation under typical environmental controls (EC1: No Spraying and No Fans; EC2: Fans; EC3: Fans and Spraying) in a naturally ventilated dairy building and assess its health risk for farm workers under three ECs.

## 2. Materials and Methods

### 2.1. Surveyed Dairy Barn and Its Environmental Controls (ECs)

A field measurement was conducted from 1 January 2021 to 31 December 2021 in a commercial dairy farm (117°04′ E, 39°25′ N) in the Beichen District, Tianjin City, Northern China. The investigated dairy barn (186 m long × 31 m wide × 4.5 m eave height) had a pitched-roof structure with ridge openings oriented west to east. Rolling curtains were equipped from top to bottom on the sidewalls to adjust the ventilation in different seasons. A total of 360 Holstein cows were housed inside the surveyed barn during the test. It was naturally ventilated with mattresses and sawdust bedding in the free stall. The farm used a total mixed ration with the corn and hay stored in a forage storage facility adjacent to the investigated barn. The cows were fed by vehicle twice a day (07:00 to 09:00 and 18:00 to 19:00), with the feed being pushed up several times during the day. Manure cleaning vehicles were used to remove dung twice a day (07:00 to 09:00 and 14:00 to 16:00). Bedding litter renewing and disinfection were operated every two days when the cows went to the milking center at 07:00, 14:00, and 22:00. The feeding and cubicle access aisles have solid floors. The lights were turned on at 17:30 and turned off at 05:30. During the measurement, the hourly mean temperature and relative humidity within the dairy barn ranged from −19.7 °C to 37.2 °C and 12% to 99.9%, respectively.

As for environmental controls (ECs), both fans and spraying systems were turned off in the winter (EC1: No Fans and No Spraying), and sidewall curtains were partially/completely closed to maintain thermal environmental comfort. In the mild seasons of spring and autumn, only mixing fans (EC2: Fans) were operated, which were usually hanging above the cubicles and headlocks to cool the dairy cows (Figure 1). In the summer, fans (24 h a day) and a spraying system (intermittently operated) were combined to alleviate heat stress (EC3: Fans and Spraying). The spraying system, installed above the headlocks of the feeding alley, was set to start once the air temperatures exceeded 25 °C. When air temperature fluctuated greatly in the months of March, June, and October, a transition between different ECs was managed, during which two or three ECs could be run alternately depending on the indoor temperature of the barns.

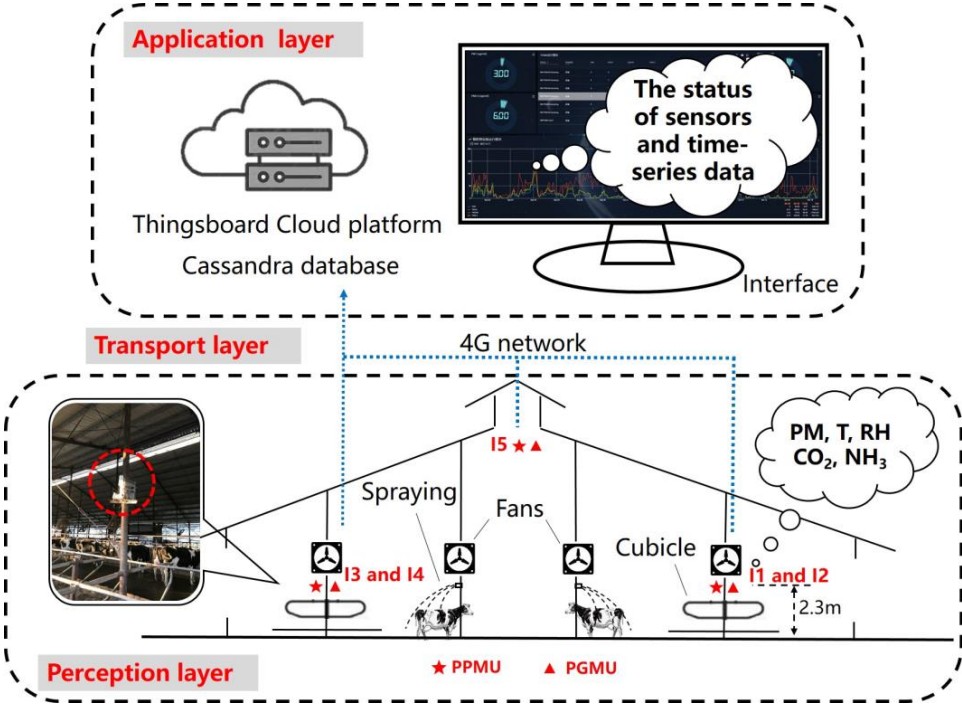

**Figure 1.** Schematics of the Internet of the Things (IoT) system framework of sensor monitoring network for dairy barns environments.

## 2.2. Environmental Parameters Measurement

### 2.2.1. Sensors Monitoring Network

A five-point sensor monitoring network based on the IoT technique was used to achieve a year-round measurement. As shown in Figure 1, the sensor monitoring network consisted of a perception layer, a transport layer, and an application layer. In the perception layer, two self-developed portable devices, including a portable particulate monitoring unit (PPMU) and portable gas monitoring unit (PGMU), were used to continuously record the $PM_{2.5}$ and TSP concentrations, as well as $CO_2$ and $NH_3$ concentrations, T, and RH, respectively, inside the dairy barn. Detailed specifications of the sensors were listed in Table 1. Each PPMU and PGMU was integrated by a microcontroller unit (MCU) and relevant sensors to detect in real-time PM and gas concentrations as well as thermal parameters inside the dairy barn. In the transport and application layers, the data were uploaded to an IoT platform (Thingsboard) via the 4G network and saved in the Cassandra database. The status of sensors (online or offline) and time-series data can be visualized on the platform to check whether the sensors were functioning properly.

**Table 1.** Technical specification of sensors integrated into portable particulate monitoring unit (PPMU) and portable gas monitoring unit (PGMU).

| Portable Unit | Parameter | Measuring Range | Resolution | Accuracy | Sensor Model and Manufacturer | Sensor Principle |
|---|---|---|---|---|---|---|
| PPMU | $PM_{2.5}$ | 0–500 $\mu g\ m^{-3}$ | 1 $\mu g\ m^{-3}$ | $\pm 10\ \mu g\ m^{-3}$ | PM5003T, Plantower, China | Light scattering |
| | TSP | 0–20 $mg\ m^{-3}$ | 1 $\mu g\ m^{-3}$ | $\pm 30\ \mu g\ m^{-3}$ | SDS198, Nova Fitness, China | |
| PGMU | T | −40–80 °C | 0.1 °C | $\pm 0.3$ °C | AM2305, Aosong Electronic, China | Electric capacity |
| | RH | 0–99.9% | 0.1% | $\pm 2\%$ | | |
| | $CO_2$ | 0–9800 $mg\ m^{-3}$ | 10 $mg\ m^{-3}$ | $\pm 98\ mg\ m^{-3}$ | MH-Z14A, Winsen, China | Infrared laser |
| | $NH_3$ | 0–38 $mg\ m^{-3}$ | 0.1 $mg\ m^{-3}$ | $\pm 0.5\ mg\ m^{-3}$ | ZEO3, Winsen, China | Electrochemistry |

During the measurement, the PPMU sampled the $PM_{2.5}$ and TSP concentrations of each point at a 1 min interval, and the average concentration was uploaded to the platform at a 5 min duration. The PGMU had a sampling cycle of 20 min. The PGMU started each cycle with a 10 min sampling duration to determine $CO_2$ and $NH_3$ concentrations, T and RH of the air sucked into the test chamber, followed by a 5-min purging period to recover the accuracy of the electrochemical $NH_3$ sensor, and a 5-min sleeping duration to release the heat generated by the embedded pump and solenoids. The PGMU data logging interval was set to 1 min, and the data was uploaded to the platform at a 20 min duration.

Sensor calibration was necessary for precise, consistent, and repeatable measurement outputs. Ahead of field measurement, the PM and gas sensors in PPMUs and PGMUs were calibrated by the manufacturer, and then the PM sensors were validated again by comparing them with tapered element oscillating microbalances (TEOMs). Their performance was also previously verified in a lab and on poultry farms [29,30]. The $CO_2$ and $NH_3$ sensors were calibrated every 3 months with calibration-grade reference gases, and the temperature and RH sensors were calibrated every 6 months with the RH Calibrator (RH CAL; Edgetech Instruments Inc., Hudson, MA, USA). During the experiment, all devices, especially PM sensors, were cleaned every two weeks, and replaced with new ones if any of a sensor's functions degenerated.

### 2.2.2. Sampling Locations

As shown in Figures 1 and 2, indoor PM, $CO_2$, $NH_3$ concentrations, T, and RH were measured by 10 samplers at 5 locations (I1 to I5) inside the barn. Four of them were placed in the center of four cubicles. The cubicles were set approximately 2.3 m above the floor (Figure 1) to monitor the air environment of the animal occupation zone without the cows' interference. The other one was close to the roof with about 1 m down the ridge opening to measure outgoing gas concentrations, and the sampling point was approximately at the



center of the barn. Ambient $PM_{2.5}$ concentrations and the air quality index were obtained from a station of the China National Environmental Monitoring Centre (CNEMC), while meteorological data, including T, RH, wind speed, and precipitation, were from a station of China Meteorological Administration (CMA). The two closest national stations were 11.4 km and 7.5 km away from the investigated dairy farm, respectively.

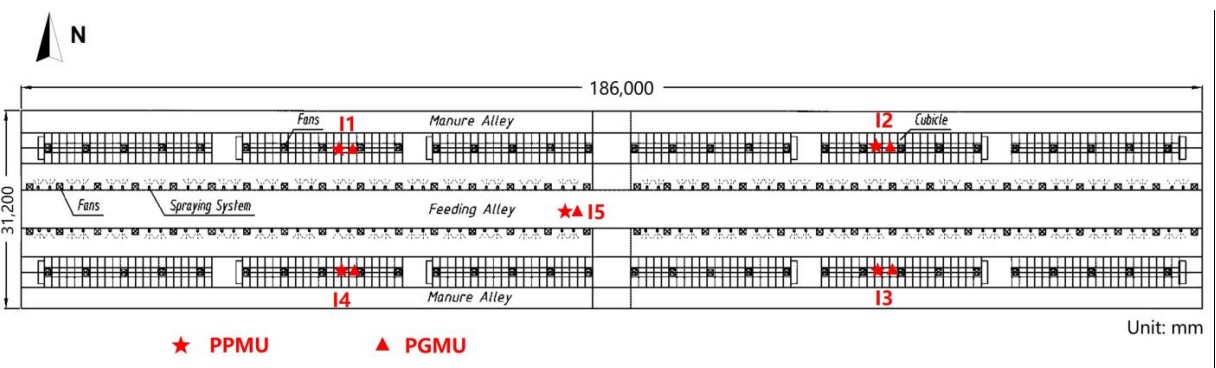

**Figure 2.** The locations of PM concentration samplers, mixing fans, and spraying systems inside the naturally ventilated dairy barn.

*2.3. Data Processing and Statistical Analysis*

In the pretreatment process, box-whisker plots were first used to remove obvious outliers, including over-range data and other data with unintended errors caused by interferences. Hourly means of all pretreated concentrations were used as the primary data for analysis. A previous study showed that ambient concentrations on smoggy (AQI > 100) days had a significant impact on indoor PM concentrations [29]. Therefore, data measured in days with AQI over 100 were excluded. Finally, pre-processed data were grouped into EC1, EC2, and EC3 to evaluate the influence of different ECs on PM concentration and variation. To ensure the accuracy of the health assessment, only PM concentration data from days with more than 20 h of valid data were used to calculate daily means.

Since the daily environmental data inside the dairy barn was non-normally distributed, the Kruskal–Wallis one-way ANOVA non-parametric test was used for statistical analysis, and the correlation between indoor TSP, $PM_{2.5}$ concentrations, and other influencers in three ECs were performed using the non-parametric Spearman's rank correlation coefficient. The significance level was 0·05. The operation time of most daily management (7:00–9:00, 14:00–16:00, 18:00–19:00) were identified as dummy variables and utilized for test purposes (referred to as Daily Management in Section 3.4). Python 3 (version 3.5) was used for data processes, analysis and plot.

**3. Results**

*3.1. Indoor TSP and $PM_{2.5}$ Concentrations in ECs*

During the year-round measurement, the annual hourly mean of indoor TSP concentration was $94.7 \pm 66.1$ μg m$^{-3}$, with a range from 4.7 to 652.1 μg m$^{-3}$, and the average indoor $PM_{2.5}$ concentration was $49.8 \pm 36.8$ μg m$^{-3}$, ranging from 1.0 to 312.3 μg m$^{-3}$. The average ambient $PM_{2.5}$ concentration was $27.9 \pm 18.6$ μg m$^{-3}$, which was nearly half of the indoor $PM_{2.5}$ level ($p < 0.01$).

Figure 3 shows the dynamic fluctuation of the hourly and daily mean of TSP and $PM_{2.5}$ concentrations under three ECs within the barn. For a better understanding of the ECs impacts, the PM data of the transitions, where two or three ECs were applied in a month, were excluded. Both the TSP and $PM_{2.5}$ concentrations fluctuated considerably under EC1, while gradually decreasing in EC2 and EC3. Generally, more pronounced fluctuations were found for TSP concentration. During the measurement, the farm purchased and stored a large amount of forage in November, lasting for a couple of days. The transportation and renewal process generated lots of particles nearby the forage storage facility, being adjacent

to the tested dairy barn on the farm, which partially explained the higher TSP and $PM_{2.5}$ concentrations of the month for the barn.

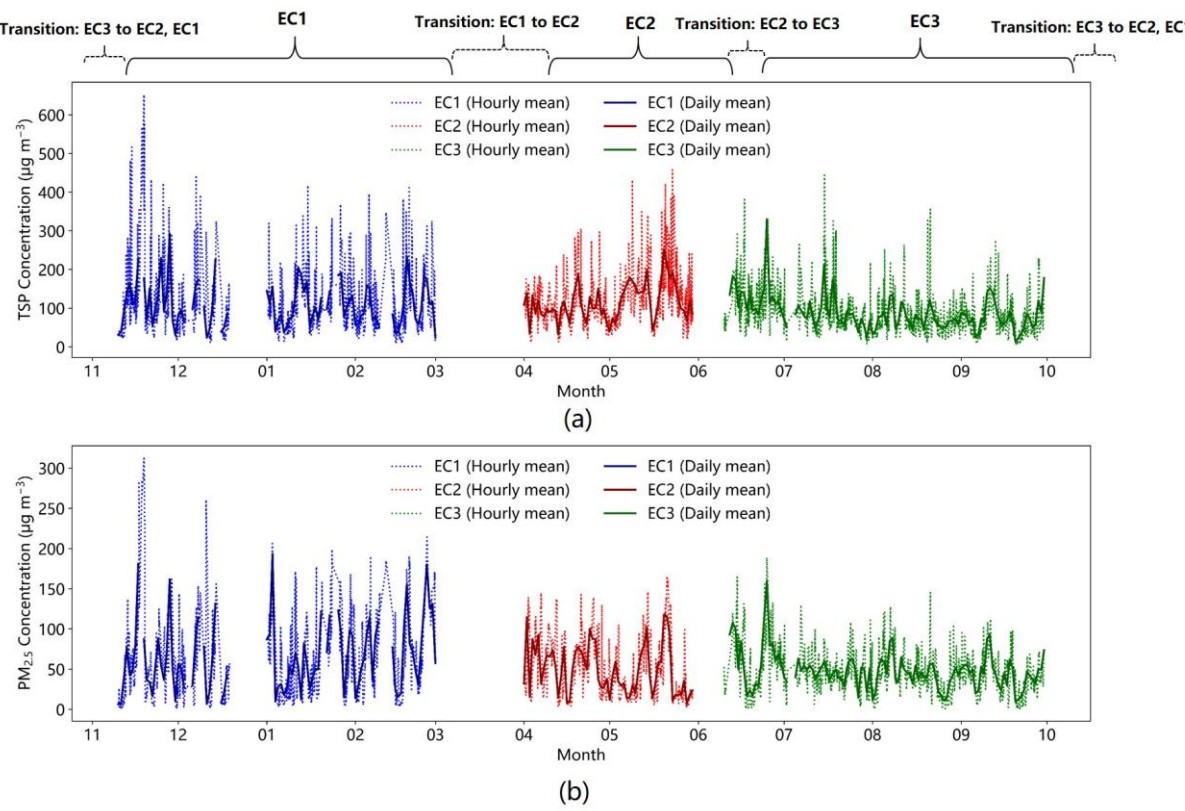

**Figure 3.** The dynamic variation of hourly and daily mean TSP (**a**) and $PM_{2.5}$ (**b**) concentrations in three ECs of the naturally ventilated dairy.

Table 2 shows that the hourly mean of TSP concentration under EC2 was higher than that under EC1 and EC3 ($p < 0.05$). Meanwhile, the hourly mean of indoor $PM_{2.5}$ concentrations under EC1 was greater than that of EC2 and EC3 ($p < 0.05$). The ambient $PM_{2.5}$ concentrations were higher in EC1 and EC2 than in EC3. The indoor RH of EC3 was significantly greater than that of EC1 and EC2 ($p < 0.05$).

**Table 2.** Hourly mean ($\pm$SD) of TSP and $PM_{2.5}$ concentrations, indoor T, and indoor RH under three ECs of the dairy barn over a year.

| Environmental Control | Indoor | | | | | | Ambient |
|---|---|---|---|---|---|---|---|
| | TSP ($\mu g\ m^{-3}$) | Range | $PM_{2.5}$ ($\mu g\ m^{-3}$) | Range | T (°C) | RH (%) | $PM_{2.5}$ ($\mu g\ m^{-3}$) |
| EC1 | 98.0 $\pm$ 75.9 [b] | 7.7–652.1 | 57.1 $\pm$ 47.2 [a] | 1.3–312.3 | 2.8 $\pm$ 6.2 [c] | 47.2 $\pm$ 18.6 [b] | 29.9 $\pm$ 20.0 [a] |
| EC2 | 116.4 $\pm$ 68.0 [a] | 12.4–457.9 | 48.3 $\pm$ 36.4 [b] | 3.2–163.8 | 18.9 $\pm$ 5.1 [b] | 48.0 $\pm$ 20.7 [b] | 28.0 $\pm$ 17.6 [a] |
| EC3 | 81.9 $\pm$ 55.0 [c] | 4.3–443.9 | 44.7 $\pm$ 25.9 [b] | 1.0–188.6 | 25.4 $\pm$ 3.9 [a] | 90.0 $\pm$ 16.6 [a] | 25.8 $\pm$ 16.9 [b] |

Note: [a–c] Different superscript lowercase letters within the same column of the same PM parameter are significantly different ($p < 0.05$).

## 3.2. Diurnal Variation of TSP and $PM_{2.5}$ Concentrations in ECs

Figure 4 depicts the weekly diurnal change of indoor TSP, $PM_{2.5}$ and ambient $PM_{2.5}$ concentrations under three ECs. The TSP concentration fluctuated clearly in EC1, with the highest values regularly appearing from 07:00 to 08:00 and 18:00 to 19:00, respectively. A similar trend, which was with less variation, can also be found for $PM_{2.5}$ concentrations. Such variations tended to be small and irregular in EC2, where TSP concentration was

constantly maintained at a high level. TSP and PM$_{2.5}$ concentrations eventually flatten out in EC3 and become stable at a lower level.

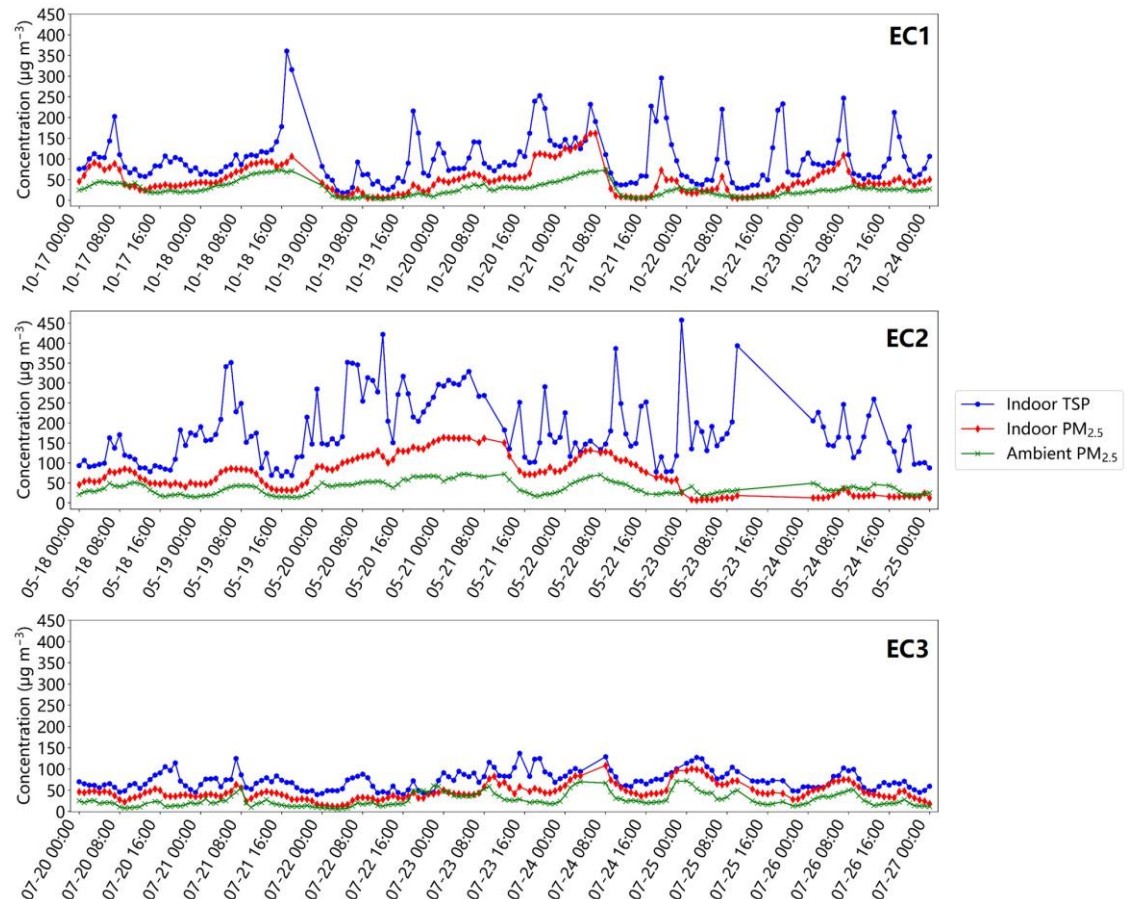

**Figure 4.** A week diurnal change of indoor TSP, PM$_{2.5}$, and ambient PM$_{2.5}$ concentrations under three ECs.

Due to the fact that the daily operations of the investigated dairy building were concentrated between 07:00 and 22:00, Table 3 presents a closer observation of TSP and PM$_{2.5}$ concentrations in daytime and nighttime under three ECs. The daytime TSP concentration was greatly higher than in nighttime in EC1 ($p < 0.05$), whereas contrary results were found in EC2 and EC3, in which higher TSP concentrations were detected in the nighttime. PM$_{2.5}$ concentrations at nighttime were higher than that of daytime in all ECs with no statistical differences observed in EC1 and EC2 ($p = 0.49$ and $p = 0.08$).

**Table 3.** Indoor TSP and PM$_{2.5}$ concentrations at daytime and nighttime in three ECs.

| Daytime and Nighttime | EC1 | | EC2 | | EC3 | |
|---|---|---|---|---|---|---|
| | TSP (µg m$^{-3}$) | PM$_{2.5}$ (µg m$^{-3}$) | TSP (µg m$^{-3}$) | PM$_{2.5}$ (µg m$^{-3}$) | TSP (µg m$^{-3}$) | PM$_{2.5}$ (µg m$^{-3}$) |
| Daytime (07:00–22:00) | 105.3 [a] | 56.5 [a] | 112.7 [a] | 47.0 [a] | 75.3 [b] | 41.7 [b] |
| Nighttime (22:00–07:00) | 82.6 [b] | 58.5 [a] | 125.0 [a] | 51.1 [a] | 94.8 [a] | 50.8 [a] |

Note: [a,b] Different superscript lowercase letters within the same column of the same PM parameter are significantly different ($p < 0.05$).

### 3.3. Correlation between Influencers and PM Concentrations in ECs

Figure 5 shows the Spearman correlation coefficients between indoor TSP and PM$_{2.5}$ concentrations and multiple influencers in three ECs. Statistically, indoor TSP, and PM$_{2.5}$

concentrations strongly correlated with ambient $PM_{2.5}$ concentration in EC1 (0.68 and 0.84, $p < 0.05$), EC2 (0.57 and 0.78, $p < 0.05$), and EC3 (0.62 and 0.86, $p < 0.05$) in a naturally ventilated barn. Indoor TSP and $PM_{2.5}$ concentrations positively correlated with daily management, such as feeding and bedding material renewing in EC1 (0.29 and 0.14, $p < 0.05$), but independent in EC2 (0.03 and 0.01, $p = 0.2$ and $p = 0.6$) and EC3 (0.05 and 0.01, $p = 0.1$ and $p = 0.7$). Both PM fractions had positive correlations with indoor $CO_2$ and $NH_3$ concentrations in three ECs. The correlations between indoor TSP, $PM_{2.5}$ concentrations and indoor T were positive and independent in EC1 (0.15 and 0.04, $p < 0.05$ and $p = 0.4$) and EC2 (0.25 and $-0.03$, $p < 0.05$ and $p = 0.3$), and negative in EC3 ($-0.14$ and $-0.16$, $p < 0.05$). RH positively correlated with indoor $PM_{2.5}$ concentration in EC1 (0.52, $p < 0.05$), EC2 (0.52, $p < 0.05$), EC3 (0.30, $p < 0.05$), as well as indoor TSP concentration in EC1 (0.40, $p < 0.05$) and EC2 (0.32, $p < 0.05$), but a weak negative correlation with TSP was found in EC3 ($-0.05$, $p < 0.05$).

As shown in Figure 5, due to the naturally ventilated system adopted for the dairy barn, there are strong positive correlations between indoor Indoor T, RH, Weather T, and RH. This explains why these variables display similar correlations with indoor PM concentrations. In three ECs, the correlation between wind speed and $PM_{2.5}$ and TSP concentrations inside the dairy barn was negative, but it was weaker in EC2 and EC3. The farm was in a typical continental monsoon climate zone, and the majority of the rainfall fell in the EC3, with much less falling in EC1 and EC2. Thus, in EC3, a negative relationship was discovered between Weath Prec and indoor TSP concentration.

### 3.4. Health Risk Assessment in ECs

The daily mean TSP and $PM_{2.5}$ concentrations under different ECs were presented in Figure 6. In this section, only data collected more than 20 h in a day were considered valid. The daily mean TSP concentrations under EC1, EC2, and EC3 were 86.9 μg m$^{-3}$, 112.1 μg m$^{-3}$, and 79.3 μg m$^{-3}$, respectively, and the corresponding indoor $PM_{2.5}$ results were 47.8 μg m$^{-3}$, 42.8 μg m$^{-3}$ and 42.7 μg m$^{-3}$. According to the thresholds of the second-level 24-h average TSP (300 μg m$^{-3}$) and $PM_{2.5}$ (75 μg m$^{-3}$) concentrations regulated by ambient air quality standards (AAQS) of China [31](red dotted lines in Figure 6), no exceeding quantity was found for daily mean TSP concentration during the year-round measurement, while the daily mean of indoor $PM_{2.5}$ concentrations, 17.9%, 11.5%, and 4.8% of measured days exceeded the 75 μg m$^{-3}$ limit in EC1, EC2, and EC3, respectively.

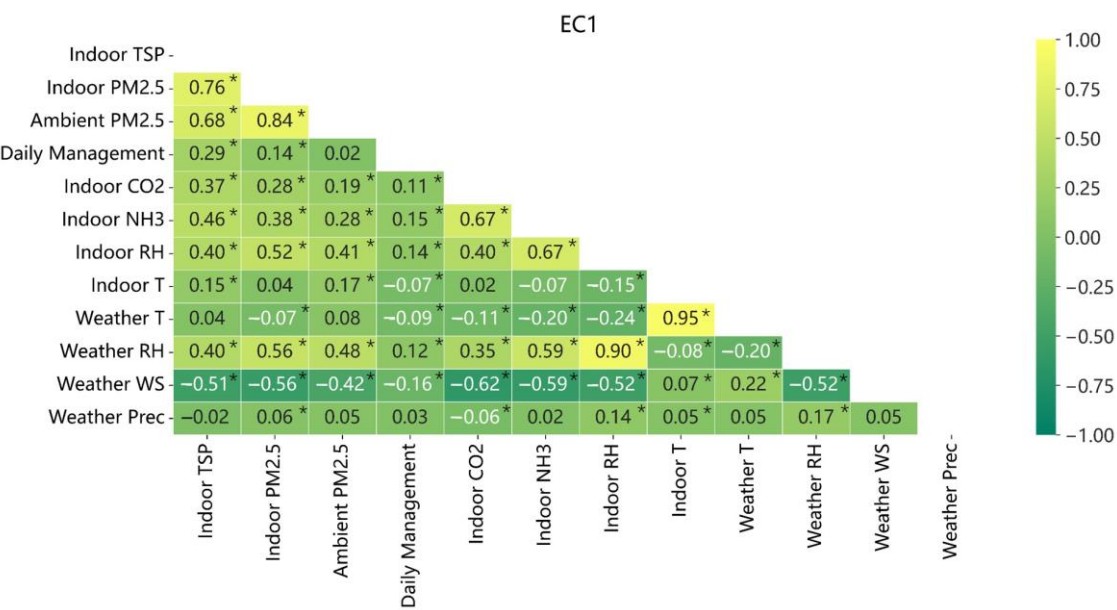

**Figure 5.** *Cont*.

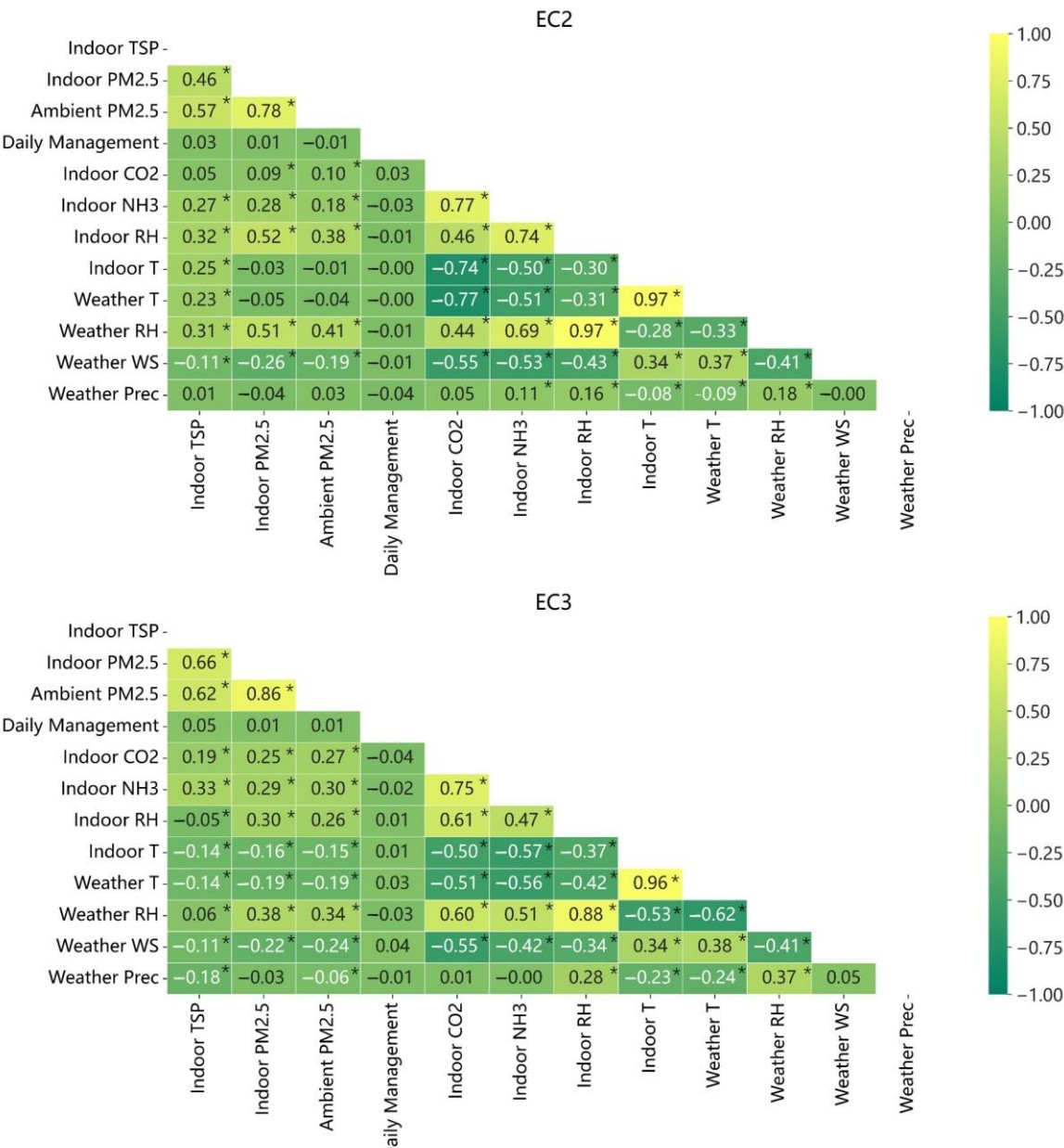

**Figure 5.** Spearman correlation coefficients between indoor TSP, $PM_{2.5}$ concentrations and multiple influencers in three ECs; Indoor T, RH, $CO_2$, $NH_3$ representing the temperature, relative humidity, carbon dioxide, and ammonia concentrations inside the dairy barn, respectively; Weather T, RH, WS, and Prec representing atmospheric temperature, relative humidity, wind speed, and precipitation data from CMA, respectively; Coefficients close to $\pm 1$ represent stronger relationships than values closer to 0, asterisk (*) is significantly different ($p < 0.05$).

Figure 7 showed the diurnal patterns of indoor TSP, $PM_{2.5}$, and ambient $PM_{2.5}$ concentrations under three ECs, which were plotted based on the measured hourly average concentration of the entire ECs periods. The TSP and $PM_{2.5}$ concentrations and variation in three ECs in Figure 7 agreed with the regularity found in Figure 4. Although there was no exceedance for hourly TSP concentration, relatively high values can be found at 07:00 to 08:00 and 18:00 and 19:00 in EC1. The hourly mean $PM_{2.5}$ concentrations exceeded 75 µg m$^{-3}$ appeared from 07:00 to 08:00, and relatively high concentrations can also be observed at night.

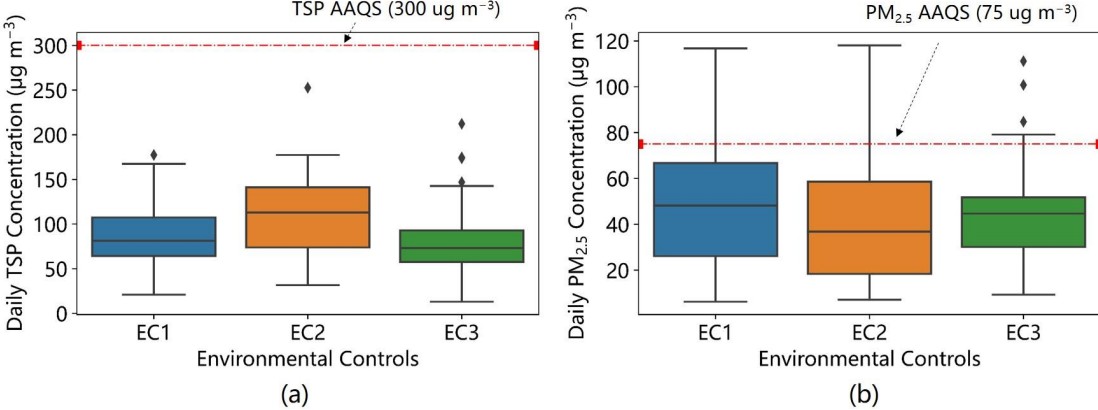
(a)          (b)

**Figure 6.** The daily mean TSP (**a**) and PM$_{2.5}$ (**b**) concentrations under three ECs; the red dotted lines representing thresholds of the second-level 24-h average TSP (300 µg m$^{-3}$) and PM$_{2.5}$ (75 µg m$^{-3}$) concentrations in the air regulated by AAQS of China standard. Rhombus presenting outliers.

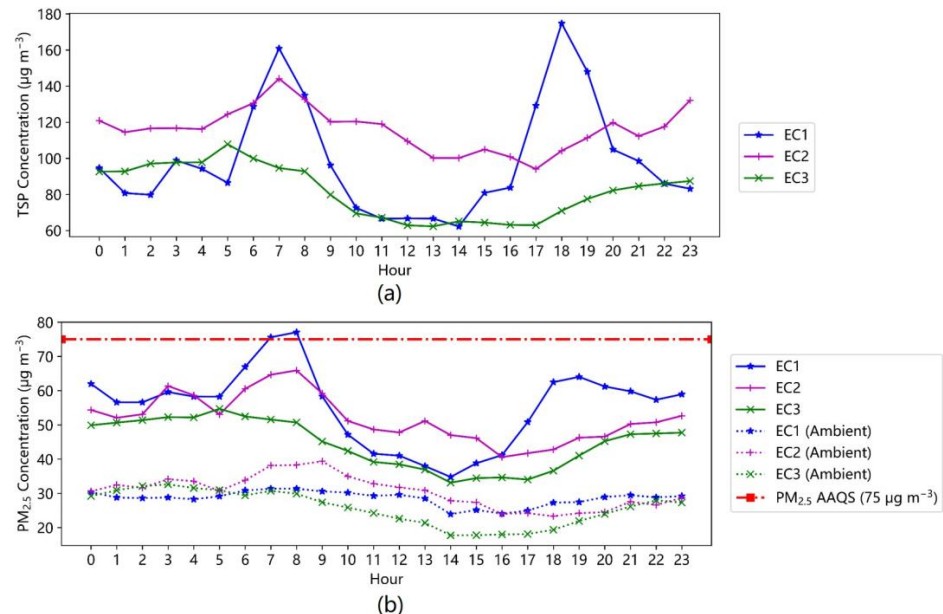

**Figure 7.** Diurnal variations of indoor TSP (**a**), PM$_{2.5}$ and ambient PM$_{2.5}$ concentrations (**b**) under three ECs; the red dotted lines in (**b**) representing thresholds of the 24-h average PM$_{2.5}$ (50 µg m$^{-3}$) concentrations in the air regulated by AAQS of China standard.

## 4. Discussion

### 4.1. PM Characteristics under Three ECs

Throughout the year-around monitoring, the hourly mean of TSP and PM$_{2.5}$ concentration in the current study were within the range detected by Kaasik and Takai's studies. The authors reported that the TSP levels in different areas of naturally ventilated barns were 125 µg m$^{-3}$ [13], and respirable dust concentrations with litter bedding in England, the Netherlands, Denmark, and Germany were 170 µg m$^{-3}$, 60 µg m$^{-3}$, 50 µg m$^{-3}$, and 30 µg m$^{-3}$, respectively [10]. Lower values have been reported in other studies [9,12,13]. The discrepancies can be mainly attributed to the type and management of bedding materials [10], measuring strategies (e.g., sampling location and frequency) [32], ventilation parameters, feeding practices, dung and slurry handling, and cow activity [14].

The EC1 was commonly used in cold seasons with the sidewall curtains partially or completely closed to maintain the thermal environment of the barn. The relatively confined environment under EC1 resulted in less air exchange and higher indoor TSP

and PM$_{2.5}$ concentrations with larger ranges. As shown in Figure 4, the PM peak occurrence in EC1 was generally agreed with the farm management, including feeding delivery, manure removal and bedding litter renewal. More cow activities, vehicles and operations during these periods consequently generated PM in the indoor air. Similar diurnal profiles of PM$_{2.5}$ and PM$_{10}$ concentrations were found in Joo and Purdy's studies [12,14]. Winkle et al. (2015) [9] also found that animal activity played an important role in spiking indoor PM$_{10}$ concentration during the day.

In EC2, fan operation increased indoor airflow, which is commonly beneficial to small particle deposition to surfaces [33] and dispersion outside of the naturally ventilated barn. Kim, Hann and Lee [21] found that the removal efficiency of PM$_{2.5}$ and PM$_{10}$ concentrations increased as fan speed increased. The proven co-effect of airflow and RH on PM particle deposition partially explained the obvious reduction of indoor PM$_{2.5}$ concentration of EC2 in comparison to EC1. Higher TSP under EC2 is probably due to the installation of mixing fans. Being installed at 15° to 25° vertical angles facing the bedding-filled cubicles and feeding alleys, mixing fans were observed to raise much bedding and feed particles into the air, which was mostly composed of coarse particles. Furthermore, high air velocity associated with the turbulence re-suspended the coarse particles that had gravitationally settled on the floor could be another reason for higher TSP in EC2. It is concluded that compared with that of EC1, in which the clear diurnal pattern was observably impacted by daily operations, because of higher air speed, the effect of daily management on PM variation with more irregular small peaks in EC2, was much compromised.

In EC3, the intermittently operated spraying system above feeding alleys dramatically raised average indoor RH to 90.0% under hot weather conditions, resulting in indoor TSP concentration considerably declining from EC2 to EC3. Augmented RH can enhance PM deposition because of the hygroscopic characteristic of particles. Under a humid environment, particles are easily agglomerated by water molecules with a large intermolecular attractive force [34], and the hygroscopic growth of particles enhances their deposition [35]. Takai et al. (1998) [10] found that high RH (>70%) contributes to low dust concentration because condensed water on the surface causes the particles to aggregate together or deposit. Furthermore, fewer airborne particles would be formed from the settled dust under the vehicle and animal activities on the moist floor or litter surface. In EC3, fan operation of 24 h a day on hot days may also enhance the PM$_{2.5}$ removal efficiency. Kim, Hann, and Lee [21] found air flow had a higher removal rate for PM$_{2.5}$ and PM$_{10}$ under humid environmental conditions. Generally, the spraying system in conjunction with continuously operating fans further weakened the fluctuations of TSP and PM$_{2.5}$, leading to a steady diurnal variation in EC3.

At nighttime, higher RH and NH$_3$ concentrations were detected due to the accumulation of manure and urine, and the PM$_{2.5}$ concentrations were higher in three ECs under a lower wind speed and less disturbance. Higher daytime TSP concentrations in EC1 were mainly owing to the daily operations, whose impact can be significantly observed in Figure 4. Since the spraying system generally operated in the daytime, TSP concentrations at night in EC3 were significantly higher than those in the daytime.

*4.2. Correlation Analysis*

Strong positive correlations between indoor TSP, PM$_{2.5}$, and ambient PM$_{2.5}$ concentrations in three ECs illustrated that atmospheric PM concentration can directly impact indoor PM because of the fully open or semi-open structure of the naturally ventilated dairy barn. Kassilk et al. (2013) [11] found PM10 and PM$_{2.5}$ concentrations inside the uninsulated loose dairy building mainly depended on the PM concentration of the outdoor air and concluded the finer the particles were, the more they were carried into the cowshed. The more stable relationship between indoor and ambient PM2.5 concentration in three ECs revealed that coarse particles were more easily affected by fans and the spraying system.

Bigger coefficients of TSP in contrast with PM$_{2.5}$ in EC1 demonstrated that daily management operation mainly contributed coarse particles inside the barn. No correlations

between daily management and indoor PM concentrations in EC2 and EC3 corroborated that the operation of fans and spraying system weakened daily management influence on PM variations, which further explained why diurnal PM patterns in EC2 and EC3 were not as clear as in EC1.

$NH_3$ is one of the important precursors of $PM_{2.5}$ because it can be carried by $PM_{2.5}$ for a long time [36,37]. Being consistent with previously reported [13,38], positive correlations between $NH_3$ and fine particle concentrations were also found in our study. $CO_2$ concentration inside the barn can indirectly represent ventilation condition, the higher the ventilation rate (lower $CO_2$ concentration), the lower the PM concentration, which explained the positive correlation between PM and $CO_2$ concentrations in three ECs.

Studies have illustrated that air temperature is not only associated with drier and more dispersed particles from feed, bedding, manure, and soil, which are more amenable to suspension in the air, but also elevated PM concentration by increasing activities of dairy cows, especially for coarse particles [9,14]. Similar relationships were also found in EC1 and EC2 in our study. However, when average temperatures in EC3 exceeded the heat stress threshold (21 °C in general) for dairy cows [39], temperature showed a negative correlation with indoor PM concentration. It was probably because high temperatures reduced animal activities [40]. One possible explanation for the positive link between indoor PM levels and RH was that portion of the liquid components (water vapor and chemicals integrated with water) were mistaken as particles by the PM sensors with the light scattering principle. Based on a similar principle, Kaasik et al. (2013) [13] also found that $PM_{2.5}$ and $PM_{1.0}$ were strongly positively correlated with the RH. Although a positive correlation between $PM_{2.5}$ concentration and RH can be found in EC3, the $PM_{2.5}$ concentrations still had a decrease from EC2 to EC3 with no statistical difference, which possibly illustrated that continuous working of fans in EC3 had more influence than RH on $PM_{2.5}$ concentrations. As opposed to the positive correlation in EC1 and EC2, the negative link between TSP concentration and RH in EC3 could illustrate that RH can lower indoor TSP levels when it exceeded a certain critical point, which was related to the hygroscopic characteristic of PM. Hygroscopic particles show significant growth at their deliquescence relative humidity [22].

External wind speed, which is proven positively correlated with the ventilation rate in the naturally ventilated dairy barn [41], can increase the removal rate of PM from the barn and vice versa [14]. Smaller correlations between wind speed and indoor PM concentrations in EC2 and EC3 suggested that fans and spraying systems abated the influence of external wind speed. Rainfall can clean the air and thus decrease the PM content [42].

*4.3. Healthy Risk Assessment under Varied ECs*

Currently, there are no TSP and PM2.5 standards for evaluating the health risks of workers inside the dairy barns. Although Donham, et al. [43] proposed the exposure concentration limits of TSP (2.4 mg m$^{-3}$) and respirable dust (0.16 mg m$^{-3}$) in terms of the health of poultry workers, the thresholds were not applicable for dairies because of the completely different indoor environmental condition. Based on dose-response relationships between environmental exposures and pulmonary function, Donham et al. also thought 2.8 mg m$^{-3}$ should be considered reasonable evidence for guidelines regarding hazardous exposure concentrations in swine facilities [44]. Due to the open design of the naturally ventilated dairy facility, the thresholds of the second-level 24-h average TSP (300 μg m$^{-3}$) and $PM_{2.5}$ (75 μg m$^{-3}$) concentrations in the air regulated by AAQS of China standard were used to assess health risk in this study. Moreover, 75 μg m$^{-3}$ was the World Health Organization (WHO) recommended value for the air quality guideline (AQG) level associated with risks to public health. Li et al. assessed the $PM_{2.5}$ concentration level and its limit exceedance of different laying hen building systems based on 75 μg m$^{-3}$ [25].

During the measurement, no exceedances were found for daily TSP concentrations in three ECs, but due to feeding operations and the high physical activity level of dairy cows after feed delivery, relatively high hourly mean TSP concentrations were found at 07:00 to 08:00 and 18:00 and 19:00. Short-term adverse responses in the lungs may occur

when workers are exposed to such conditions. Compared with EC1, the steadily high concentration of TSP in EC2 should also be taken seriously. Brunekreef and Forsberg [45] concluded that admissions for respiratory disorders typically rose by 2–6% for every 10 µg m$^{-3}$ increase in coarse particulate concentration. The highest effect size exceeded 10% per 10 µg m$^{-3}$, particularly with longer averaging times [46]. The hourly mean TSP concentration in EC2 was about 20 ug m$^{-3}$ and 30 ug m$^{-3}$ higher than EC1 and EC3.

Exceeding daily PM$_{2.5}$ concentrations in three ECs suggested its higher environmental risk than TSP. Compared to other sizes of PM, the particles with diameters between 2.1 and 4.7 mm were reported to contain the most microorganisms, easily leading to inflammation [47]. Overall, higher PM$_{2.5}$ concentration in the test barn mainly occurred during the daily management and at night, especially in EC1. In EC1, with the exception of the hours between 09.00 and 17.00 when part of the sidewall curtains and doors were opened on warm days, the PM$_{2.5}$ concentration was always kept at a relatively high level. Long-term exposure to such an environment may cause farm workers to suffer from respiratory function impairment. Pope III, et al. [48] assessed the health risk of long-term exposure to air pollution and found each 10 µg m$^{-3}$ elevation in fine particulate resulted in approximately 4%, 6%, and 8% increased risk of all-cause, cardiopulmonary, and lung cancer mortality, respectively. As the ventilation decreased, the hourly mean PM$_{2.5}$ concentration increased by around 10 µg m$^{-3}$ from EC3 to EC1. It can also be observed that nighttime PM$_{2.5}$ concentrations in three ECs were higher than in the daytime. Furthermore, the nighttime PM$_{2.5}$ concentration in EC3 was near 10 ug m$^{-3}$ higher than daytime. Arteaga found dairy facilities had higher endotoxin in PM2.5 when compared with the control facility [49]. Therefore, it is suggested that protection procedures should be taken when farmers work inside the barn, and more health care should be provided to farmers under EC1 conditions, especially during the daily management time.

## 5. Conclusions

Based on a year-round continuous monitoring with five representative points inside a naturally ventilated dairy barn, the dynamics of PM concentration and its relationship with multiple influencers under EC1 (No Fans and No Spraying), EC2 (Fans), and EC3 (Fans and Spraying) were characterized. During the measurement, the hourly mean TSP and PM$_{2.5}$ concentrations were 94.7 µg m$^{-3}$ and 49.8 µg m$^{-3}$, respectively. The TSP concentration in EC2 was much higher than that in EC1 and EC3. Meanwhile, the PM$_{2.5}$ concentrations in EC1 were the highest, followed by the EC2 and EC3. TSP and PM$_{2.5}$ concentrations fluctuated during the daily operations at 07:00 to 08:00 and 18:00 to 19:00 in EC1, while relatively steady diurnal patterns in EC2 and EC3 were found. The correlation between daily management, external wind speed, and indoor PM concentration decreased when fans and spraying systems were operated in EC2 and EC3. No exceedances were found for daily TSP concentrations in three ECs, but the ratios that the daily mean PM$_{2.5}$ concentrations exceeded the 75 µg m$^{-3}$ limit of the air reached 17.9%, 11.5%, and 4.8% in EC1, EC2, and EC3, respectively. The exceedance of PM$_{2.5}$ concentrations mainly occurred during the daily operation period and at night, In conclusion, fans and spraying systems impacted PM concentration and variation greatly, for TSP in particular, prevention measures should be taken for farm workers when working inside the barn, especially in EC1.

**Author Contributions:** Conceptualization, Y.L. (Yujian Lu), Z.S. and C.W.; writing—original draft preparation, Y.L. (Yujian Lu); writing—review and editing, X.Y., C.L. and C.W.; data acquisition and calibration, Y.L. (Yongzhen Li), Z.F. and L.E. All authors have read and agreed to the published version of the manuscript.

**Funding:** This research was supported by the National Natural Science Foundation of China (31972614) and the earmarked fund of the China Agriculture Research System (CARS36).

**Data Availability Statement:** Data will be made available on reasonable request from the corresponding author.

**Acknowledgments:** The authors are grateful to Jinri jiankang dairy farm for providing a field experimental site.

**Conflicts of Interest:** The authors declare that they have no known competing financial interests or personal relationships that could have appeared to influence the work reported in this paper.

## Nomenclature

| | | | |
|---|---|---|---|
| ECs | Environmental controls | EC1 | EC with No Fans and No Spraying |
| PM | Particulate matter | EC2 | EC with Fans |
| $PM_{2.5}$ | Particulate matter with an aerodynamic diameter of up to 2.5 μm | EC3 | EC with Fans and Spraying |
| TSP | Total suspended particulate. Particulate matter with an aerodynamic diameter of up to 100 μm | PPMU | Portable particulate monitoring unit |
| T | Temperature | PGMU | Portable gas monitoring unit |
| RH | Relatively humidity | AQI | Air quality index |
| $CO_2$ | Carbon dioxide | MCU | Microcontroller unit |
| $NH_3$ | Ammonia | CNEMC | China National Environmental Monitoring Centre |
| SD | Standard deviation | CMA | China Meteorological Administration |
| IoT | Internet of Things | AAQS | Ambient air quality standards |

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
