# Peer review of "Characterizing a Year-Round Particulate Matter Concentration and Variation under Different Environmental Controls in a Naturally Ventilated Dairy Barn"

_agriculture, doi:10.3390/agriculture13040902_

Round 1

Reviewer 1 Report

Authors did a systematic work on the measurement of particulate matter concentrations on a dairy farm. Dust levels are good references for environment regulators to think about regulations or laws for protecting environment and ecosystems. Some suggestions for authors' consideration:

1) the ratio of PM2.5 to TSP is too high (>0.5), suggest authors to present PM10 data as well.

2) what about emissions of PM from the farm. Dairy house dust levels are not big concerns in most time. Emissions information is critical for developing mitigation strategies. 

3) Any bedding material information (sand?) and what moisture content? 

4) Do you have the data collected in coldest time of the year such as Dec., Jan, or Feb? PM levels in coldest time reflects worst air quality due to lowest ventilation. 

5) Line 259: provide p value if it is larger than 0.05.

Generally, the writing is well organized. 

Authors did a systematic work on the measurement of particulate matter concentrations on a dairy farm. Dust levels are good references for environment regulators to think about regulations or laws for protecting environment and ecosystems. Some suggestions for authors' consideration:

1) the ratio of PM2.5 to TSP is too high (>0.5), suggest authors to present PM10 data as well.

2) what about emissions of PM from the farm. Dairy house dust levels are not big concerns in most time. Emissions information is critical for developing mitigation strategies. 

3) Any bedding material information (sand?) and what moisture content? 

4) Do you have the data collected in coldest time of the year such as Dec., Jan, or Feb? PM levels in coldest time reflects worst air quality due to lowest ventilation. 

5) Line 259: provide p value if it is larger than 0.05.

Generally, the writing is well organized. 

Author Response

Thanks for your constructive comments that have helped us to improve our manuscript. The point-by-point response to the comments is given below (revised sentences were marked as blue):

Reviewer 2 Report

Manuscript agriculture-2345843 entitled “Characterizing a Year-Round Particulate Matter Concentration and Variation under Different Environmental Controls in a Naturally Ventilated Dairy Barn. Please notice the following:

General view: The manuscript discussed a good point of view in moderate language and grammar. The manuscript must be subjected to copyediting and further simplification of the used expressions, as well as a necessity to update the references.

The manuscript could be accepted for publication after MINOR revision.

Title: Long and preferred to be modified into “A Year-Round Particulate Matter Concerning Environmental Controls in a Naturally Ventilated Dairy Barn”.

Abstract: Clear and informative. Simplification of the sentences and rephrasing has to be carried out to enhance readability and understanding.

Keywords: Rearrange the keywords in alphabetical order.

Introduction: The introduction has to be rearranged into three paragraphs i.e., 1. Introduction 2. Significance of the study, and 3. Aim of the study. A certain degree of rephrasing has to be carried out to enhance readability and understanding.

The aim: Concise, clear, and informative.

Materials and methods: Please notice the following:

1.      Provide information on microclimatic conditions for the 360 Holstein cow understudy in their barn such as temperature and relative humidity, ventilation system, lighting system, feeding & watering systems, drainage system, rodent control measures, pest control measures, wild bird control measures, and cleaning and disinfection procedures.

2.      Provide the statistical model used in the analysis.

3.      A certain degree of rephrasing has to be carried out to enhance readability and understanding.

Results: Avoid rewriting the values as the results have to be expressed in your own words rather than using the numbers from the tables all over. Other than that, the results are novel. Simplification of the sentences and rephrasing has to be carried out to enhance readability and understanding.   

Discussion: Clear to a certain extent and contributing to knowledge with a good degree of comparison and speculation. Avoid citing the tables and figures all over in the discussion section and reduce the usage of numbers to simplify the text for better understating and readability.

Conclusion: reduce the size of the conclusion for better understanding and readability.

Authors’ contributions: Not listed.

Funding: Clear and informative.

Acknowledgment: Not listed.

References: MUST BE UPDATED as 24.4% (12 out of 49) were published in the past five years. The percentage has to increase to at least 30-40%. Old references give indications that the study is no longer a case of importance.

Tables: Well organized and presented.

Figures: Well organized and presented.

The manuscript must be subjected to copyediting and further simplification of the used expressions.

Author Response

Thanks for your constructive comments that have helped us to improve our manuscript. The point-by-point response to the comments is given below (revised sentences were marked as blue)
